# Stronger influence of anthropogenic disturbance than climate change on century-scale compositional changes in northern forests

Victor Danneyrolles[1,2,3], Sébastien Dupuis[1], Gabriel Fortin[1], Marie Leroyer[1], André de Römer[1], Raphaële Terrail[1], Mark Vellend[3,4], Yan Boucher[3,5], Jason Laflamme [3,6], Yves Bergeron [2,3] & Dominique Arseneault [1,3]

Predicting future ecosystem dynamics depends critically on an improved understanding of how disturbances and climate change have driven long-term ecological changes in the past. Here we assembled a dataset of >100,000 tree species lists from the 19th century across a broad region (>130,000km$^2$) in temperate eastern Canada, as well as recent forest inventories, to test the effects of changes in anthropogenic disturbance, temperature and moisture on forest dynamics. We evaluate changes in forest composition using four indices quantifying the affinities of co-occurring tree species with temperature, drought, light and disturbance. Land-use driven shifts favouring more disturbance-adapted tree species are far stronger than any effects ascribable to climate change, although the responses of species to disturbance are correlated with their expected responses to climate change. As such, anthropogenic and natural disturbances are expected to have large direct effects on forests and also indirect effects via altered responses to future climate change.

[1] Département de biologie, chimie et géographie, Université du Québec à Rimouski, Rimouski, QC G5L 3A1, Canada. [2] Chaire industrielle CRSNG-UQAT-UQAM en Aménagement Forestier Durable, Université du Québec en Abitibi-Témiscamingue, Rouyn-Noranda, QC J9X 5E4, Canada. [3] Centre d'étude de la forêt (CEF), Montréal, QC H2X 1Y4, Canada. [4] Département de Biologie, Université de Sherbrooke, Sherbrooke, QC J1K 2R1, Canada. [5] Direction de la recherche forestière, Ministère des Forêts, de la Faune et des Parcs, Québec, QC G1P 3W8, Canada. [6] Direction des inventaires forestiers, Ministère des Forêts, de la Faune et des Parcs, Québec, QC G1H 6R1, Canada. Correspondence and requests for materials should be addressed to V.D. (email: victor.danneyrolles@uqat.ca)

Anthropogenic disturbance and climate change are two of the major drivers of terrestrial ecosystem dynamics worldwide[1,2]. Assessing the relative importance of these and other factors in causing past ecological change is of critical importance to predicting future ecological dynamics[3,4], but empirical tests face several challenges. First, the time scale of available historical data might constrain our ability to detect the effects of particular global change drivers. For example, some studies that cover the past 40–50 years capture a period of more pronounced warming than land-use change and thus are perhaps more likely to detect climate change impacts[5–7]. Second, changes that are ascribed to one driver such as climate warming (e.g., elevational distribution shifts) might in fact, upon closer examination, be more likely explained by a different factor such as change in land use[8–10]. This is especially problematic for the warming of the twentieth century, which has been paralleled by major changes in anthropogenic disturbance regimes[11], such that these two drivers of ecosystem dynamics can mask or modify each other's effects. Finally, the influence of "climate" is often assessed by focusing only on temperature, while changes in precipitation can have equally large ecological impacts[1,2]. Clearly, there is a pressing need for explicit empirical tests of the influence of changes in temperature, precipitation and disturbance regimes on ecological communities across broad spatial and temporal scales.

Forests make major contributions to global biodiversity and ecosystem services and in some regions land surveys conducted during the nineteenth century provide unique opportunities to assess changes in forest composition over a period of time representing particularly drastic changes in both land use and climate[12]. Climate change affects tree recruitment, growth and mortality, and thus potentially forest composition and species' geographic ranges[13–15]. However, the response of tree communities to climate change can be delayed[16] or catalyzed[17] due to demographic and dispersal processes, and to the simultaneous impacts of competition, land use and disturbance[17–19]. Anthropogenic disturbances imposed by land uses come in different forms (e.g., agricultural clearing, logging, anthropogenic fires), which collectively influence forest dynamics by conferring advantages to disturbance-adapted, early-successional, short-lived and fast-growing tree species, which over a period of decades or centuries are successively replaced by late-successional, long-lived, shade-tolerant species[20–22]. An increasing number of simulation-based studies have explored potentially interacting effects of disturbance and climate change on forest dynamics[17,18], but it remains difficult to empirically test model predictions over relevant time frames[23]. Based on studies of pre-settlement and contemporary forest composition in eastern North America, a lively—and as yet unresolved—debate has arisen concerning the relative importance of disturbances vs. climate in driving regional-scale forest changes in recent centuries[23–26].

Here we quantify changes in tree functional composition over the last century across a >130,000 km$^2$ region in eastern Canada, at the ecotone between temperate and boreal forests (Fig. 1a), and test the relative importance of climate change and anthropogenic disturbance as drivers of these changes. Over this time period, this region has experienced substantial changes in temperature and moisture (Fig. 1b, c), and profound land-use changes linked to European settlement (i.e., rapid increase in disturbance rates due to the expansion of agriculture and industrial forestry[27–30]). We present a large dataset of early land survey observations (>100,000 taxa lists recorded between 1790 and 1900; Supplementary Fig. 1) and modern forest inventories conducted during 1980–2010, both aggregated to a 25 km$^2$ grid (Supplementary Fig. 2). Under the hypothesis that climate change has been an important driver of forest dynamics, we test the prediction of a

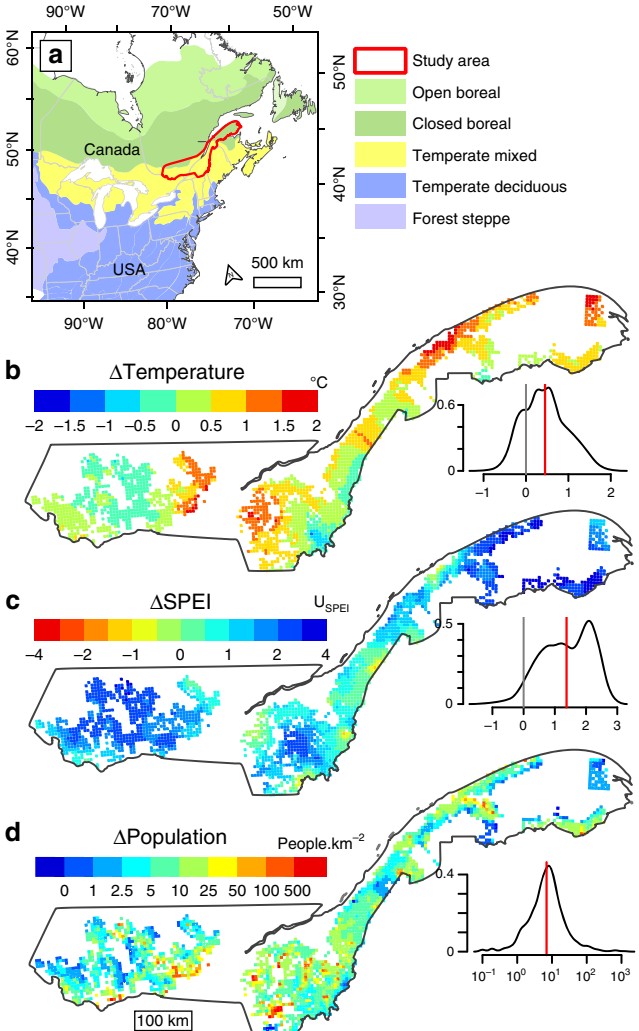

**Fig. 1** Changes in temperature, moisture and population. Location of the study area (**a**), and changes from 1901 to 1980 in mean annual temperature (ΔTemperature; **b**) and in the standardized precipitation evapotranspiration index (ΔSPEI; **c**) and changes in population density since 1831 (ΔPopulation; **d**); see Methods for a description of data sources. SPEI is based on the difference between monthly precipitation and potential evapotranspiration, and negative values of ΔSPEI indicate an increase in drought, while positive values indicate an increase in moisture. Change in population density was calculated as the difference between population density in 1831 and the highest population density recorded since the end of the nineteenth century. Smoothed distributions are shown next to each map (vertical gray and red lines show 0 and mean values, respectively, and ΔPopulation was log-transformed)

shift in forest composition toward more warm- and moisture-adapted tree species[24,25], and a positive correlation of the magnitude of such changes with the magnitude of change in temperature and moisture. Similarly, under the hypothesis that anthropogenic disturbance has driven forest change, we test the prediction of a shift toward more disturbance-adapted species, correlated with areas of more intense human influences. We quantified functional community composition using four indices (calculated using independent data) that represent the average affinities of co-occurring taxa with temperature (community temperature index (CTI)), drought (community drought tolerance index (CDTI)), light (community shade tolerance index (CSTI)) and disturbance (community disturbance index (CDI)). The latter two indices are expected to respond to disturbance

negatively and positively, respectively. Changes in climate-related community indices were then tested for statistical relationships with changes from 1901 to 1980 in mean annual temperature (ΔTemperature, Fig. 1b) and in the standardized precipitation evapotranspiration index (ΔSPEI, a moisture index based on the balance between precipitation and potential evapotranspiration; Fig. 1c). Changes in disturbance-related community indices were tested against changes in human population density since the early nineteenth century (ΔPopulation; Fig. 1d), the latter being a good proxy for historical changes in land-use intensity. Overall, we show an overriding influence of anthropogenic disturbances compared to climate change in driving tree community dynamics since the nineteenth century, with disturbance-related changes potentially slowing or accelerating future community shifts in response to changes in temperature and precipitation.

## Results

**Changes in functional composition.** Before calculating community-level functional composition indices, scores across taxa were standardized to a common scale (0–1; see Methods, Supplementary Tables 1 and 2), such that the magnitudes of change in the four indices (Fig. 2) can be directly compared. The dominant temporal change was a shift toward more disturbance-adapted communities, with 92% of cells (25 km$^2$) showing an increase in the community disturbance index (mean ΔCDI = 0.13; Fig. 2d). In comparison, we found clear but weaker trends toward communities composed of less shade-tolerant and less drought-tolerant species: 77% of cells showed a decrease in the community shade tolerance index (mean ΔCSTI was −0.06; Fig. 2c) and 77% of cells also showed a decrease in the community drought tolerance index (mean ΔCDTI = −0.05; Fig. 2b). In contrast, the community temperature index (ΔCTI) showed no clear trend on average, with a mean ΔCTI of 0.01 and high variance, including some grid cells with either strong decreases or increases (Fig. 2a).

**Correlation with changes in predictor variables.** Linear mixed models showed that change in population density (ΔPopulation) had strong and significant effects—in the predicted directions—on both ΔCSTI ($P < 0.001$; Fig. 3c) and ΔCDI ($P < 0.001$; Fig. 3d). Conversely, while climatic predictors had significant effects on ΔCTI and ΔCDTI, the effects were relatively weak and opposite in direction to those predicted. Linear mixed models showed that the relationship between ΔTemperature and ΔCTI was significant only when controlling for changes in disturbance-related indices (i.e., ΔCDI and ΔCSTI; $P = 0.011$; Fig. 3a), with greater temperature increases associated with reductions in CTI. ΔCDTI tended to be higher (less negative) in areas with greater increases in the SPEI (Fig. 3b), and this relationship was non-significant when controlling for changes in disturbance-related indices (i.e., ΔCDI and ΔCSTI; $P = 0.123$, Fig. 3b).

## Discussion

These results point to a dominant role of anthropogenic disturbances underlying forest compositional changes during recent centuries in these northern temperate forests. Disturbance-related adaptations of different taxa were captured in two indices, one focused on well-established shade tolerance scores, CSTI[31,32], thought to capture expected responses to severe disturbances (e.g., clear-cut logging, anthropogenic fire). The other index (CDI) was developed specifically for this study to capture expected responses to a wider range of disturbance severities (e.g., selective logging; see Methods, Supplementary Table 1). This CDI showed the strongest temporal changes, involving increases for a broad spectrum of disturbance-adapted taxa (e.g., early-

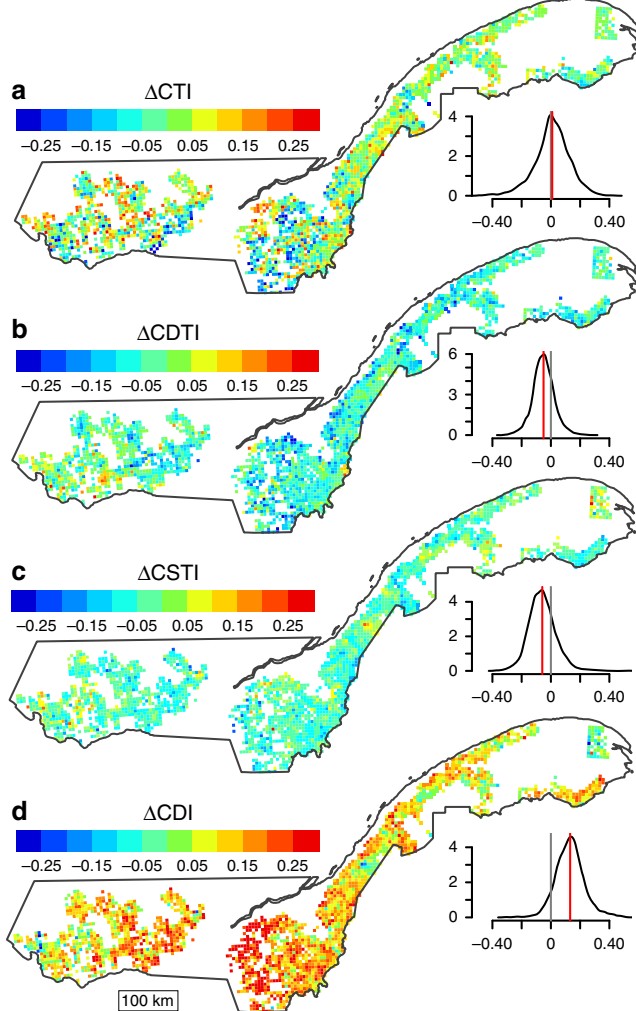

**Fig. 2** Changes in functional composition. Changes in the community temperature index (ΔCTI; **a**), the community drought tolerance index (ΔCDTI; **b**), the community shade tolerance index (ΔCSTI; **c**) and the community disturbance index (ΔCDI; **d**) between 1790–1900 and 1980–2010. Smoothed distributions are shown next to each map (vertical gray and red lines show 0 and mean values, respectively). Detailed maps of community indices in each period are shown in Supplementary Figure 3

successional *Populus* spp. and *Betula papyrifera*, as well as more shade-tolerant but logging-favored *Acer* spp. and *Abies balsamea*) at the expense of long-lived late-successional taxa (*Picea* spp., *Tsuga canadensis*, *Fagus grandifolia*; Supplementary Figure 4). The CSTI showed less pronounced changes, mostly linked to increases in early-successional deciduous taxa (*Populus* spp., *Betula papyrifera*; Supplementary Figure 4). These results are consistent with similar land-use-driven shifts toward more disturbance-adapted and less shade-tolerant taxa in recent centuries found for portions of northeastern North America further to the south[23,33,34].

Our results provide no clear evidence of direct climate change impacts upon the composition of these forests over the twentieth century. Changes in the moisture regime can significantly influence forest dynamics, particularly through severe drought[14,32,35,36], and as predicted our results showed a shift toward less drought-tolerant communities. However, the magnitude of such changes was not correlated with changes in moisture, and the shift in the average CDTI may have been a correlate of disturbance-driven changes. Taxon-level affinities with

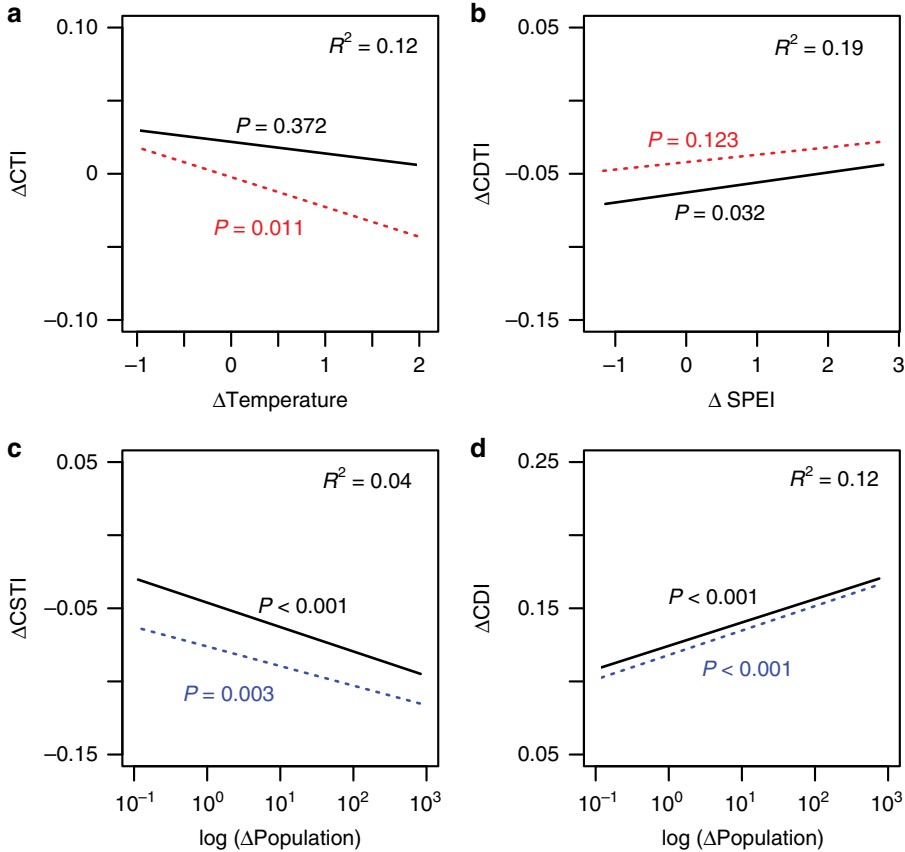

**Fig. 3** Relationships between changes in functional composition and potential predictors. Linear mixed models: **a** ΔCTI with ΔTemperature, **b** ΔCDTI with ΔSPEI and **c**, **d** ΔCSTI and ΔCDI with log-transformed ΔPopulation. Black lines and text show effects estimated by models with single predictors, while colored lines and text show effects after controlling for changes in other indices (see Methods). Red lines and text show effects after controlling for changes in disturbance-related indices (ΔCSTI and ΔCDI), while blue lines show effects after controlling for changes in climate-related indices (ΔCTI and ΔCDTI). P values are shown for each line and the conditional $R^2$ values of single-predictor models are shown in the upper right corners. CTI community temperature index, CDTI community drought tolerance index, SPEI standardized precipitation evapotranspiration index, CSTI community shade tolerance index, CDI community disturbance index

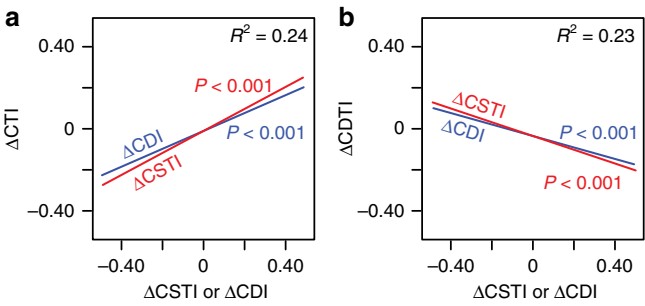

**Fig. 4** Relationships between changes in different community indices. Linear mixed models: **a** ΔCTI against ΔCSTI and ΔCDI, and **b** ΔCDTI against ΔCSTI and ΔCDI. P values are shown for each line and the conditional $R^2$ values are shown in the upper right corners. CTI community temperature index, CDTI community drought tolerance index, CSTI community shade tolerance index, CDI community disturbance index

disturbance, light and drought were correlated to different degrees (Supplementary Table 3), and changes in CDTI were significantly affected by changes in the disturbance-related indices CSTI and CDI (Fig. 4b). ΔCDI and ΔCSTI were both negatively correlated with ΔCDTI (Fig. 4), mainly due to increases of the disturbance-adapted and drought-intolerant *Abies balsamea*[37,38] at the expense of drought-tolerant, late-successional taxa such as

*Betula alleghaniensis*, *Picea* spp. and *Thuja occidentalis* (Supplementary Figure 4). We found no average change in the CTI, and site-to-site variation in ΔCTI also appears to have been influenced by disturbance-related changes (Fig. 4a). Increased CTI was associated with increased CDI due to increases of disturbance- and warm-adapted *Acer* spp. and decreases of late-successional, cold-adapted *Picea* spp. (Supplementary Figure 4). Decreased CTI was associated with decreased CSTI due to increases of early-successional, cold-adapted *Populus* spp. and *Betula papyrifera*, with decreases at some sites of the warmest adapted, late-successional taxa (i.e., *Tsuga canadensis*, *Betula alleghaniensis*, *Fagus grandifolia*; Supplementary Figure 4). Overall, we detected no signal of warming-related changes, and while we cannot totally exclude some influence of an altered moisture regime on vegetation dynamics[24,25], it is clear that any effects of climate change (altered temperature and/or rainfall) appear to have been dwarfed by the dominant influence of disturbance.

Natural disturbances such as wildfire or insect outbreaks, the frequency and intensity of which have increased in many parts of the world as an indirect effect of climate change[39,40], could also have played a role in the observed shift toward more disturbance-adapted tree communities. We are, however, confident in attributing past forest changes to the direct effects of anthropogenic disturbance rather than indirect effects of climate change via increased natural disturbance. First, fire frequency has actually declined in this region due to increased moisture[41] as well as

efficient fire suppression since the mid-twentieth century[42]. Second, while insect outbreaks have occurred cyclically during the twentieth century (principally *Choristoneura fumiferana*[43] and *Malacosoma disstria*[44]), there is no indication of an increased outbreak frequency[43]. In addition, the tree species most affected by these defoliators (*Abies balsamea*, *Populus tremuloides*, *Acer saccharum*) are among the species that actually showed the strongest increases over time in our study area (Supplementary Figure 2). These considerations, along with the significant correlation between change in population density and both ΔCSTI and ΔCDI, provide clear evidence of an overriding influence of land-use-driven disturbances on long-term changes in the composition of forests in southern Québec.

Our results have important implications for understanding forest dynamics in an era of rapid global changes. First, changes in successional processes induced by disturbance represent more powerful drivers of forest dynamics compared to direct effect of climate change at the temporal scale of a century. Given ongoing changes in disturbance regimes that can be mediated by both land use[45,46] and climate change[39,40], we might expect that forest disturbance will significantly influence future responses to changes in temperature and moisture regime[47,48]. We found evidence of disturbance-induced compositional changes that countered the direction of expected warming-induced dynamics (e.g., increases in less warm-adapted taxa) despite the clear trend of increasing temperatures during the last century[49]. To the extent that these changes were driven by anthropogenic disturbance, this may help explain the frequently observed lags in responses of tree communities to recent warming[16] and it suggests that disturbance may hinder forest ecosystems' adaptation to the predicted temperature increase in the coming century[50]. At the same time, we found some disturbance impacts that appear to favor the same taxa expected to increase with greater moisture (e.g., increases in less drought-adapted taxa), thus potentially accelerating this aspect of climate-induced changes. However, the shift we observed towards less drought-tolerant tree communities took place during one of the wettest periods of the last five centuries in northeastern North America[24,25], and if future droughts increase in intensity and frequency[51], disturbance-related changes may exacerbate tree mortality and compromise forest resilience[14]. More generally, our study highlights how large-scale secondary succession and recovery processes induced by land use or other sources of disturbance, which may persist for decades to centuries[48], could significantly influence the response of ecosystems to future climate change.

## Methods

**Tree community data**. Preindustrial forest composition was characterized using early land survey data aggregated to a 25 km$^2$ grid across southern Québec. Early land survey data were extracted from logbooks reporting the original surveys of 302 townships between 1790 and 1900 (Supplementary Figure 1). Surveys were conducted along the boundaries of the townships and subdividing range lines within townships (every 1.6 km, Supplementary Figure 1), where surveyors described the forest composition in the form of lists of taxa (e.g., hemlock, beech and maples) (note that these data were collected in a different way than the General Land Office data widely used to reconstruct past forests characteristics in the northeastern United States[52]). In total, 103,011 lists of taxa from 1790 to 1900 were georeferenced using historical and modern digital cadastral maps and used in this study (Supplementary Figure 1). These observations were divided into two geometric types: point observations (69% of total observations, usually spaced 100–200 m apart) and line descriptions (31% of total observations and usually pertaining to one lot boundary of ~260 m). In order to incorporate these two data types in the same database, a weight was assigned to each observation that represented the length of line observations and the mean spacing of point observations (the mean distance to the previous and next observations[53]). Data were then aggregated to a 25 km$^2$ grid; in each grid cell we estimated a taxon's frequency (Supplementary Figure 2) as the cumulative weight of observations in which the taxon was mentioned divided by the total weight of observations. A previous study validated the high accuracy of this method to estimate the nineteenth-century

forest composition by comparison with the results obtained using early forest inventories available for a restricted portion of the study area[54].

Modern forest composition was quantified using the Québec government's forest inventories since 1980. These inventories are based on 400 m$^2$ circular plots distributed across different types of productive forest through stratified random sampling (Supplementary Figure 1). Only plots that were located within a maximum distance of 3 km from historical observations were retained, resulting in a total of 59,359 plots. The vast majority of plots have been continuously forested during the last century, with <1% of plots having developed on abandoned farmlands. Within these plots, all stems >9 cm diameter at breast height of each species were measured and inventoried and were used to compute species basal area per plot. Surveyors of the nineteenth century did not systematically distinguish all phylogenetically close species and thus several species had to be grouped at the genus level to match the taxa that were recorded by historical surveyors (*Picea* spp., *Pinus* spp., *Acer* spp., *Populus* spp.). Because the nineteenth-century surveyors only specified the most abundant taxa in their observations, we also controlled the number of taxa per modern plot in order to obtain more comparable datasets. Taxa that represented <5% of the total basal area of a modern plot were removed; this threshold was chosen to obtain roughly the same number of taxa per observation in both historical and modern datasets (Supplementary Figure 1). Modern taxa frequencies were finally aggregated to the 25 km$^2$ grid by calculating number of plots where a taxon is present divided by the number of plots in the corresponding grid cell (Supplementary Figure 2).

**Taxa and community indices**. Changes in functional composition were quantified using indices that characterize the affinities of taxa within communities with temperature (CTI), drought (CDTI), light (CSTI) and disturbance (CDI). These community indices were calculated as follows. First, indices that quantify taxon-specific relationships with temperature, drought tolerance, shade tolerance and disturbance were assigned to the main 17 taxa present in the historical and modern datasets (Supplementary Tables 1 and 2). The "taxon temperature index" (TTI) was calculated as the median annual temperature across the geographic range of a given taxon[7,15,55] using interpolated annual surface temperature maps[56] and continental scale distribution maps[57]. To assess the robustness of our analyses to the decision to use median temperatures, we also calculated variants of TTI using the mean, 10th percentile, and 90th percentile of temperatures across the geographic range of a given taxon (see Supplementary Methods; Supplementary Tables 4 and 5), with the latter two representing the cold and warm limits of distributions, respectively. Because none of these variants significantly changed the results (Supplementary Figure 7 and Supplementary Table 6), we report only results for the median in the main text. Taxon drought tolerance and shade tolerance indices (TDTI and TSTI, respectively, Supplementary Tables 1 and 2) were taken from a database of northern hemisphere trees and shrubs[31,32]. These indices were derived from a meta-analysis of published literature ranking taxa according to their drought and shade tolerance on a common scale (from low tolerance; 1, to high tolerance; 5); they have been frequently used to quantify tree functional composition[32,58]. The rankings obtained with our taxon climatic indices (TTI and TDTI) are also consistent with recent analyses of growth–climate relationships of restricted pools of taxa considered in our analysis[37,38]. Concerning disturbance-related indices, since TSTI mostly accounts for adaptions to severe stand-replacing disturbances (e.g., land clearing, clearcutting), we also developed for this study a more comprehensive taxon disturbance index (TDI) that considers adaptations to both severe and partial disturbances (e.g., partial cutting). TDI was calculated with a point system involving 11 key traits (Supplementary Tables 1) taken from various sources including the PLANTS and TRY databases[59,60] and references on tree species ecology[61,62]. Five traits were considered primarily as adaptations to stand-replacing disturbances: shade tolerance, growth rate, longevity, age at sexual maturity and capacity of vegetative reproduction. Six additional traits were considered to reflect adaptations to both stand-replacing and partial disturbance: fruiting frequency, seed abundance, effective seed dispersal, germination substrate requirement, seedling vigor, and response to release. For each trait, the range of values across species was first standardized to vary from 0 to 1 (giving them equal weight) and then summed across traits for a given taxon to obtain TDI values.

For species grouped at the genus level (*Picea* spp., *Pinus* spp., *Acer* spp. and *Populus* spp.), values of TTI, TDTI, TSTI and TDI were averaged across the main species present in study area (our results were robust to various combinations of species included in these taxon-level values; see Supplementary Methods; Supplementary Tables 7 and 8, Supplementary Figures 8 and 9). For each time period separately, community indices were calculated for each 25 km$^2$ grid cell as the weighted mean index value across taxa found in that grid cell, with weights determined by taxon frequency. In order to obtain values comparable across the four community-level indices, despite the different scales of taxon-level indices, all taxon-level indices were first standardized to vary from 0 to 1 among taxa prior to calculations of community-level indices (Supplementary Table 2). Thus, each community index potentially ranges from 0 to 1 (Supplementary Figure 3), and changes in community indices (the difference between the 1980–2010 and 1790–1900 periods) potentially range between −1 and 1.

**Climate and population density data**. Climate data were obtained at the 25 km$^2$ grid scale for the 1901–1980 period using BIOSIM 10 software[63]. The year 1901

corresponds to the earliest date for which accurate climate records are available in our study area, while 1980 correspond to the earliest modern inventories used in the analysis. Changes in mean annual temperature were calculated as the linear slope of mean annual temperature per calendar year for each cell separately. To assess changes in moisture regime, we used the SPEI, which is based on the difference between monthly precipitation and temperature-induced potential evapotranspiration[64]. Negative values of SPEI indicate dry events, with values < −1.5 commonly considered as severe drought, while positive values indicate wet events. The 24-month SPEI between January 1901 and December 1980 were calculated in R using the SPEI package[65] and changes in SPEI were calculated with the linear slope of SPEI per month for each cell separately.

In the absence of precise and extensive data on land-use history over the relevant spatial and temporal scales in our study area, we used changes in human population density since the early nineteenth century as a proxy for the intensity of anthropogenic disturbances over this time period. Sub-regions of our study area experienced their most intense periods of land use at different times. For example, some areas reached their peak in agricultural development and rural population as early as the end of the nineteenth century[28], shortly after surveyors opened the territory for settlement (Supplementary Figures 1 and 5). Other regions reached such peaks between the middle and the end the twentieth century[29]. In order to account for this heterogeneity, we computed changes in mean population density between 1831 and, for each cell separately, the maximum mean population density recorded in one of the three subsequent censuses in 1871, 1951 or 2001 (see details in Supplementary Figure 5), which represent three key periods of land-use history in our study area[28,29]. Data from the 1831 and 1871 censuses have been made available by the Centre Interuniversitaire d'Études Québecoises (CIEQ; Projet GÉORIA, version 2003) and data from the 1951 and 2001 censuses are freely available on the Canada's Century Research Infrastructure (CCRI) website (https://ccri.library.ualberta.ca).

**Statistical analysis**. Relationships between changes in community indices and potential predictors were tested with linear mixed effects models, which were constructed in R with the nlme package[66]. In all models, we accounted for natural biogeographic variation in forest composition by first conducting a spatially constrained clustering of 25 km² cells based on preindustrial composition and then using the six resulting groups (see Supplementary Figure 6) as a random factor in all models. Spatial autocorrelation was taken into account within cluster groups using an exponential spatial correlation structure constructed with the corExp function contained in the nlme package[66]. We first tested for individual effects of potential predictors on community indices: we tested effects of ΔTemperature on ΔCTI, of ΔSPEI on ΔCDTI and of ΔPopulation on both ΔCSTI and ΔCDI. Second, because taxon-level affinities with disturbance, light and drought were correlated to some degree (Supplementary Table 3), we tested effects of potential predictors after controlling for changes in other community indices[32]. Effects of ΔTemperature on ΔCTI and of ΔSPEI on ΔCDTI were tested after accounting for changes in disturbance-related indices (i.e., ΔCSTI and ΔCDI). Conversely, effects of changes in population density on ΔCSTI and ΔCDI were tested after accounting for changes in climate change-related indices (i.e., ΔCTI and ΔCDTI). Finally, we tested correlations between changes in climate- and disturbance-related community indices, with ΔCTI or ΔCDTI as dependent variables (separately) and ΔCSTI and ΔCDI as predictors.

## Code availability
The code used to fit mixed models can be made available from the corresponding author upon request.

## Data availability

The historical data that support the findings of this study are available from the Laboratoire d'Écologie Historique et de Dendrochronologie (dominique_arseneault@uqar.ca) upon reasonable request. Modern forest data are freely available from the Ministère des Forêts, de la Faune et des Parcs du Québec website (https://mffp.gouv.qc.ca/le-ministere/acces-aux-donnees-gratuites/).

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

## Acknowledgements

The authors thank Jean Noël and Marie-Claude Lambert for their help with the modern forest inventories and climate data and Byron Moldofsky, Laurent Richard and Marc St-Hilaire for their help with historical population density data. This project was financially supported by the Ministère des Forêts, de la Faune et des Parcs du Québec (MFFP, project #142332085), the Fonds de Recherche du Québec Nature et Technologies (FRQNT), the Natural Sciences and Engineering Research Council of Canada (NSERC), the Chaire de recherche sur la forêt habitée (UQAR) and through a MITACS-sponsored industrial partnership with the forest product company Rayonier Advanced Materials (La Sarre, QC).

## Author contributions

V.D. and D.A. designed the study and methodology with substantial inputs from M.V., Y. Boucher and J.L. S.D., G.F., M.L. A.d.R., R.T. and V.D. extracted and compiled the historical data. V.D. analyzed the data. V.D. wrote the first draft with substantial inputs from D.A., M.V., Y. Boucher, J.L. and Y. Bergeron.

## Additional information

**Competing interests:** The authors declare no competing interests.

