## [Peer Review File · Nature Communications]

Reviewers' comments:

Reviewer #1 (Remarks to the Author):

The manuscript by Danneyrolles and co-workers analyzes forest compositional changes over roughly a century for a large area of land in southern Quebec, Canada. The authors subsequently aim to attribute these changes to climate and land-use, regressing changes in functional composition against indicators of climate and land-use change. The main finding of the authors is that land-use change had a stronger influence on changes in community composition than climate change.

A particular strength of the study is its unique historical dataset, based on historical land surveys. Utilizing these historical sources – although being limited in terms of how and what data were recorded – allows the authors to study compositional changes consistently over large spatial and temporal scales. Also, the analysis is highly timely, as it is important to clearly distinguish the signal of climate change from other drivers (such as land-use change).

Reading the manuscript for the first time, one thing that confused me, starting from the title and going all the way through the introduction, was the terminology the authors use. When talking about disturbances, they actually refer to human disturbances (such as management), and not natural disturbances (such as wildfire). In order to avoid this confusion for the reader I suggest rewording to “land-use” whenever you refer to human disturbances. This will make the text much more clear.

A main issue that I see is in disentangling and attributing vegetation changes to underlying drivers. First of all, the indices of functional community composition used here are naturally correlated. This is also acknowledged by the authors (cf. Table S3), but how this inherent correlation structure of traits is accounted for in the analysis remains unclear to me. This is important though, as it might be that a change in a composition index reported here is simply the result of the correlation structure in the traits considered. For instance, as we know that light and water demand are correlated (e.g., Niinemets and Valladares 2006, *Ecol. Monographs* 76), the changes in light demand interpreted here as response to land-use might simply follow from tree responses to changing water availability (or vice versa!). It is thus important to account for these correlations and show that any changes in the traits (and indices derived from them) are actually going beyond what would be expected simply from trait correlations.

A second concern that I have is that the authors only consider direct climate change effects (i.e., changes in temperature and precipitation) and disregard indirect effects of climate change. Climate change could, for instance, increase natural disturbances such as wildfires and insect outbreaks. This is a trend that has been reported for many parts of Canada (e.g., Kurz et al. 2008 *PNAS* 105, Wotton et al. 2010, *International Journal of Wildland Fire*), and affects large parts of the globe (Kautz et al. 2017, *Global Change Biology*, Seidl et al. 2017 *Nature Climate Change*). These would then – in turn – have a similar signature in your analysis as land-use changes. So in order to maintain your rather strong statement that climate change was not an important driver of the observed vegetation dynamics you will need to control for the indirect effects of climate change, especially for changes in natural disturbance regimes.

Furthermore, I find the temperature index used here rather weak, and suspect that some of the insensitivity of the results might stem from the particular way this index was derived. Put simply, after 20 years of species distribution modeling we know quite well that the median temperature across the realized niche of a species is not a very good indicator for its thermal requirements. More details on this are in the comments below.

For a journal like *Nature Communications*, that caters to a broad, interdisciplinary readership, the title and abstract would have to be revised strongly. As it is, the main message was hard to get

even for me. Non-specialists will almost certainly not be able to understand the (potential) significance of your work from the current abstract and title. Also, the citation style needs to be revised. For example, in many instances reference numbers are not given as superscript.

More detailed comments on the text:

l25-27: Unclear – couldn't changing disturbance be an (indirect) effect of climate change?

l32: none of the references given here explicitly reports on disturbances – maybe consider adding some of the recent disturbance and climate change literature

l35-38: I don't understand this sentence – please rephrase for clarity

l57: have explored

l58-59: I don't agree here – particularly the simulation-based studies are usually conducted over quite extended time horizons. Are you rather talking about empirical studies here? If so, the first part of the sentence is misleading...

l84-86: I doubt that contemporary population density is indeed the best proxy for land-use. For one thing, many rural (= sparsely settled) areas are subject to heavy industrial forestry while the vicinity of densely populated metropolitan areas are often managed for recreation (= lower land-use impact). For another thing, settlement density changed considerably over time, and decreased again in many parts as settlers moved west (at least in the eastern US), making the current population density not a good measure of the accumulated human impact over the last centuries.

l112: what does the 30 in parenthesis mean after CSTI?

l150-151: how can you be sure that the successional dynamics that you are observing is indeed in response to human disturbances, and not the long-term forest recovery after natural disturbances that happened before your study period?

l152: as far as I can see your analysis only covers 79 years (1901 - 1980) and not multiple centuries, correct?

l201: is the genus-level sufficient to conduct a functional traits analysis? The genus *Acer*, for instance, is rather broad and contains both species that are shade-tolerant and species that are light-demanding. Averaging across all species of the genus thus clearly doesn't make sense...

l216-217: this assumes that the median of a species' realized niche is a good indicator for its temperature demands/ limits. Given all the advances in species distribution modeling over the last two decades, I find this a rather naive approach. What about extreme temperatures, for instance? What about differences between the fundamental and realized niche of a species?

Reviewer #2 (Remarks to the Author):

The paper by Danneyrolles et al. deals with a globally important topic, namely the consideration of the climate effects and those by anthropogenic disturbances on/in forest ecosystems. There are already studies on natural disturbances and their effect on forests in comparison to climate change, e.g. Seidl et al. (2017 in *Nature Climate Change*), but the study presented here introduces as a new approach anthropogenic disturbance over a very long period of time. The extremely extensive data set, which covers a very large area and extends over a century, together with the fact that the area is located in the ecotone between the temperate and boreal zones, undoubtedly gives the study a very high status in the scientific community. As far as I know, it is the first paper of its kind (in this research field) that covers a very large region and centuries of time combined with quite high-resolution data.

In the introduction, convincing arguments for the study are presented and the result takes into account the claim to make it clear that anthropogenic disturbances have far stronger effects on tree communities than the significant climate changes that have taken place in the region. This will certainly encourage further similar studies.

The entire paper is characterised by a very clear and comprehensible structure, above all I find the illustrations very successful, in order to make clear to the reader how it was done and which results came out. For example, Figure S5a is relatively small and it is therefore not easy to see whether 5 or 6 clusters is actually the best number of clusters, but together with Figure S5c a

convincing image is given. I therefore enjoyed reading the paper very much. I see no ambiguities or errors in the methodological approaches, including statistical analysis, and find the discussion points presented convincingly.

Nevertheless, there is one major point of criticism from my side. I disagree with the authors that there is no more comprehensive data on the history of land use. In any case, there is also a considerable discrepancy between the complex calculation of the TDI (see lines 228 to 236) and the very rough estimate of anthropogenic disturbance (see lines 258-261).

I do not come from Canada, but I have a lot of experience in researching historical maps, data and the like, and for Canada there is an atlas

"Atlas of Canada"; published by the Department of the Interior, Canada. Honourable Frank Oliver, Minister, 1906. Prepared under the direction of James White F.R.G.S., Geographer

With or in addition individual map(s)

„Canada population density 1901. Ontario and Quebec. WHITE - 1906“ and “Density of population, 1901. Maritime Provinces and Quebec”, printed 1960“

In addition, there is a link with which you can display the human population density back to about 1850:

<http://mercator.geog.utoronto.ca/Georia/mapbox-hacolp/Website/densitymap.html>

It is therefore not clear to me why these data or sources have not been used to use population density change as a proxy for land-use intensity/history change.

Moreover, with the help of such data the statement in lines 66 to 67 should be substantiated, otherwise it remains a statement without a reference, which represents a certain disproportion to the well-prepared other evidences of all other changes (climate, Taxa community) represented in the area.

Apart from that I have a few little things to note

1) There is widespread knowledge that the Canadian forests are continually affected by fire, insects and disease. Thus quite a lot of different natural disturbances are active. It should therefore become clear in the title as well as in the text (not only in line 37) that anthropogenic disturbances are meant here. This will also help to avoid misunderstandings regarding the cited literature. In other words, it will then become clear why literature that takes natural disturbances into account, such as that by Seidl et al. (2017), was not quoted.

2) In most Figures capital letters have been used to identify the individual Figures, but lower case letters are indicated in the text of the Figure captions: that should become consistent.

Reviewer #3 (Remarks to the Author):

I have reviewed the paper "Dominant impact of disturbance relative to climate change on forest compositional changes over the past century in eastern Canada" by Dr. Danneyrolles and colleagues. This is an excellent piece of work on all levels and I very much support the conclusions. My only concern is to what extent is it novel compared to their previous papers on the study area and subject. I have read these papers but have not gone back to do a detailed analysis to see how this paper differs from the others. Maybe it is by investigating climate change as a driver. I do know that these papers contain information about forest dynamics and the disturbance, land-use drivers. If the handling editor is convinced that the contribution of this paper is substantially different than the other I am fully supportive of it being published as is.

RESPONSE TO REVIEWERS COMMENTS

Reviewer #1:

The manuscript by Danneyrolles and co-workers analyzes forest compositional changes over roughly a century for a large area of land in southern Quebec, Canada. The authors subsequently aim to attribute these changes to climate and land-use, regressing changes in functional composition against indicators of climate and land-use change. The main finding of the authors is that land-use change had a stronger influence on changes in community composition than climate change.

A particular strength of the study is its unique historical dataset, based on historical land surveys. Utilizing these historical sources – although being limited in terms of how and what data were recorded – allows the authors to study compositional changes consistently over large spatial and temporal scales. Also, the analysis is highly timely, as it is important to clearly distinguish the signal of climate change from other drivers (such as land-use change).

Reading the manuscript for the first time, one thing that confused me, starting from the title and going all the way through the introduction, was the terminology the authors use. When talking about disturbances, they actually refer to human disturbances (such as management), and not natural disturbances (such as wildfire). In order to avoid this confusion for the reader I suggest rewording to “land-use” whenever you refer to human disturbances. This will make the text much clearer.

We changed the term “disturbance” to “anthropogenic disturbance” in the title and in *Abstract* (lines 21-24), to clarify that we are referring specifically to land-use related disturbance. We prefer using "anthropogenic disturbances" rather than "land-use change" because the vast majority of the modern plots used in this study have had continuous forest cover over the relevant time period, subject to extensive logging but not to transformations to other land-use types such as agriculture (lines 210-211 in the *Methods*), even if the broader region has experienced such land-use transformations.

*A main issue that I see is in disentangling and attributing vegetation changes to underlying drivers. First of all, the indices of functional community composition used here are naturally correlated. This is also acknowledged by the authors (cf. Table S3), but how this inherent correlation structure of traits is accounted for in the analysis remains unclear to me. This is important though, as it might be that a change in a composition index reported here is simply the result of the correlation structure in the traits considered. For instance, as we know that light and water demand are correlated (e.g., Niinemets and Valladares 2006, *Ecol. Monographs* 76), the changes in light demand interpreted here as response to land-use might simply follow from tree responses to changing water availability (or vice versa!). It is thus important to account for these correlations and show that any changes in the traits (and indices derived from them) are actually going beyond what would be expected simply from trait correlations.*

We were perhaps not sufficiently clear in communicating that our mixed model analysis was especially designed to resolve this problem (results presented in Figures 3 and 4, see *Methods*; lines 296-312), so we appreciate this prompting to clarify the issue. We first tested for individual effects of potential predictors on changes in community indices (e.g. Δ SPEI on Δ CDTI). Second,

specifically to account for correlations between changes in community indices, we tested effects of the potential predictors on changes in community indices after correcting for changes in other community indices (e.g., Δ SPEI on Δ CDTI when controlling for Δ CSTI and Δ CDI). Since our analyses confirmed the dominant role of anthropogenic disturbance in vegetation changes (i.e., significant effects Δ Population upon Δ CSTI and Δ CDI while no conclusive effects of Δ Temperature and Δ SEPI on Δ CTI and Δ CDTI, respectively), we finally tested the effects of changes in disturbance-related community indices on climate-related community indices. Overall, we feel that this statistical approach allowed us to conduct strong tests disentangling and attributing vegetation changes to underlying drivers. However, we acknowledge that the earlier version of our manuscript could have had clearer explanations and justifications for this statistical approach, so we have added some text for clarity (see lines 304-307).

A second concern that I have is that the authors only consider direct climate change effects (i.e., changes in temperature and precipitation) and disregard indirect effects of climate change. Climate change could, for instance, increase natural disturbances such as wildfires and insect outbreaks. This is a trend that has been reported for many parts of Canada (e.g., Kurz et al. 2008 PNAS 105, Wotton et al. 2010, International Journal of Wildland Fire), and affects large parts of the globe (Kautz et al. 2017, Global Change Biology, Seidl et al. 2017 Nature Climate Change). These would then – in turn – have a similar signature in your analysis as land-use changes. So in order to maintain your rather strong statement that climate change was not an important driver of the observed vegetation dynamics you will need to control for the indirect effects of climate change, especially for changes in natural disturbance regimes.

We added a whole paragraph discussing the potential role of changes to the natural disturbance regime caused by climate change (lines 150-163; with reference to recent literature; Seidl et al. 2017; *Nature Climate Change*, Sommerfeld et al. 2018; *Nature Communications*). In this new paragraph, we point out that in our study area, the impact of natural disturbances on compositional changes has most likely been small relative to the impact of anthropogenic disturbance, and in particular (citing relevant regional studies) that climate change has not increased the frequency of fire. In other words, our statistical signal of disturbance impacts is not an indirect consequence of climate change. Nevertheless, we used this enlargement of the discussion to comment on the broader, forward-looking context, noting “ongoing changes in disturbance regime that can be mediated by both land-use and climate change” (lines 167-168; 182-184). We think that these additions clarify and broaden the scope of the manuscript.

Furthermore, I find the temperature index used here rather weak, and suspect that some of the insensitivity of the results might stem from the particular way this index was derived. Put simply, after 20 years of species distribution modeling we know quite well that the median temperature across the realized niche of a species is not a very good indicator for its thermal requirements. More details on this are in the comments below.

This is a fair criticism, which we appreciate. As such, we went back to the original distribution and climate data in order to extract alternative indices of species temperature affinities, including both warm and cold extremes. Sensitivity tests of our results to these alternatives are presented in *Supplementary information 2* section (lines 605-705), and they clearly demonstrate the robustness of our results and conclusions.

For a journal like Nature Communications, that caters to a broad, interdisciplinary readership, the title and abstract would have to be revised strongly. As it is, the main message was hard to get even for me. Non-specialists will almost certainly not be able to understand the (potential) significance of your work from the current abstract and title. Also, the citation style needs to be revised. For example, in many instances reference numbers are not given as superscript.

We broadened and clarified our title, from “Dominant impact of disturbance relative to climate change on forest compositional changes over the past century in eastern Canada” to “Anthropogenic disturbance is more important than climate change in explaining compositional changes in northern forests over the past century”. We also made several changes in the abstract in order to clarify our main message (lines 19-30).

*Lines 25-27: Unclear – couldn’t changing disturbance be an (indirect) effect of climate change?
l32: none of the references given here explicitly reports on disturbances – maybe consider adding some of the recent disturbance and climate change literature.*

Response to this concern have been developed above.

Lines 35-38: I don’t understand this sentence – please rephrase for clarity.

We have rephrased this sentence (lines 35-38).

Line 57: have explored.

This change has been made (line 60).

Lines 58-59: I don’t agree here – particularly the simulation-based studies are usually conducted over quite extended time horizons. Are you rather talking about empirical studies here? If so, the first part of the sentence is misleading...

We clarified this sentence (Lines 59-62).

Lines 84-86: I doubt that contemporary population density is indeed the best proxy for land-use. For one thing, many rural (= sparsely settled) areas are subject to heavy industrial forestry while the vicinity of densely populated metropolitan areas are often managed for recreation (= lower land-use impact). For another thing, settlement density changed considerably over time, and decreased again in many parts as settlers moved west (at least in the eastern US), making the current population density not a good measure of the accumulated human impact over the last centuries.

As suggested by reviewer #2, we now use historical human population density data instead of only modern population density in order to calculate a proxy for land-use intensity in our analyses. Because different portions of our study area experienced their peak in land-use intensity and population density at different times, we computed changes in mean population density, for each 25km² cell separately, between 1831 and the year during which maximum mean population density was subsequently recorded in that cell. Using this new predictor variable, we obtained results qualitatively identical and quantitatively very similar to those in the earlier version of the manuscript. We report the details and justification for these new population data and analyses in the *Methods* section (lines 280-294) and in Figure S1.5 (lines 575-587).

Obviously, the use of these new data is also reflected in several parts of the main text (e.g., *Introduction* section; lines 89-91, Fig. 1 caption; lines 485-493).

Line 112: what does the 30 in parenthesis mean after CSTI?

This was a typographical error in citation format that has been corrected (line 117).

Lines 150-151: how can you be sure that the successional dynamics that you are observing is indeed in response to human disturbances, and not the long-term forest recovery after natural disturbances that happened before your study period?

We consider this hypothesis as very unlikely because of the large size of our study area, and the time frame involved. Long-term recovery after a disturbance would lead to an increase in shade tolerant and less disturbance-adapted taxa, whereas we found precisely the opposite. The reverse pattern may be possible in a few specific localities, due to specific edaphic/ecological conditions, but not at the regional scale, where the only plausible explanation is anthropogenic disturbance that occurred between the early and modern surveys.

Line 152: as far as I can see your analysis only covers 79 years (1901 - 1980) and not multiple centuries, correct?

80 years is indeed the minimum time span between preindustrial (1790-1900) and modern observations (1980-2010). However, most of the study area have been surveyed before 1860-1870 (see Figure S1.1, lines 517-528).

Line 201: is the genus-level sufficient to conduct a functional traits analysis? The genus Acer, for instance, is rather broad and contains both species that are shade-tolerant and species that are light-demanding. Averaging across all species of the genus thus clearly doesn't make sense... +

We designed and now report sensitivity tests, which demonstrate the robustness of our results to different possible decisions underlying these analyses. These additional results are presented in the *Supplementary information 2* section (lines 605-705), to which we refer in the main text *Methods* section (lines 232-237 and 258-260). In short, our results were clearly robust to alternative methodological choices here.

Lines 216-217: this assumes that the median of a species' realized niche is a good indicator for its temperature demands/ limits. Given all the advances in species distribution modeling over the last two decades, I find this a rather naïve approach. What about extreme temperatures, for instance? What about differences between the fundamental and realized niche of a species?

Response to this concern have been developed above.

Reviewer #2:

The paper by Danneyrolles et al. deals with a globally important topic, namely the consideration of the climate effects and those by anthropogenic disturbances on/in forest ecosystems. There are already studies on natural disturbances and their effect on forests in comparison to climate change, e.g. Seidl et al. (2017 in Nature Climate Change), but the study presented here

introduces as a new approach anthropogenic disturbance over a very long period of time. The extremely extensive data set, which covers a very large area and extends over a century, together with the fact that the area is located in the ecotone between the temperate and boreal zones, undoubtedly gives the study a very high status in the scientific community. As far as I know, it is the first paper of its kind (in this research field) that covers a very large region and centuries of time combined with quite high-resolution data.

In the introduction, convincing arguments for the study are presented and the result takes into account the claim to make it clear that anthropogenic disturbances have far stronger effects on tree communities than the significant climate changes that have taken place in the region. This will certainly encourage further similar studies. The entire paper is characterised by a very clear and comprehensible structure, above all I find the illustrations very successful, in order to make clear to the reader how it was done and which results came out. For example, Figure S5a is relatively small and it is therefore not easy to see whether 5 or 6 clusters is actually the best number of clusters, but together with Figure S5c a convincing image is given. I therefore enjoyed reading the paper very much. I see no ambiguities or errors in the methodological approaches, including statistical analysis, and find the discussion points presented convincingly.

There is one major point of criticism from my side. I disagree with the authors that there is no more comprehensive data on the history of land use. In any case, there is also a considerable discrepancy between the complex calculation of the TDI (see lines 228 to 236) and the very rough estimate of anthropogenic disturbance (see lines 258-261). I do not come from Canada, but I have a lot of experience in researching historical maps, data and the like, and for Canada there is an atlas "Atlas of Canada"; published by the Department of the Interior, Canada. Honourable Frank Oliver, Minister, 1906. Prepared under the direction of James White F.R.G.S., Geographer With or in addition individual map(s), Canada population density 1901. Ontario and Quebec. WHITE - 1906" and "Density of population, 1901. Maritime Provinces and Quebec', printed 1960". In addition, there is a link with which you can display the human population density back to about 1850: [link]. It is therefore not clear to me why these data or sources have not been used to use population density change as a proxy for land-use intensity/history change.

*As suggested, we now use historical human population density data instead of only modern population density in order to calculate a proxy for land-use intensity in our analyses. Because different portions of our study area experienced their peak in land-use intensity and population density at different times, we computed changes in mean population density, for each 25km² cell separately, between 1831 and the year during which maximum mean population density was subsequently recorded in that cell. Using this new predictor variable, we obtained results qualitatively identical and quantitatively very similar to those in the earlier version of the manuscript. We report the details and justification for these new population data and analyses in the *Methods* section (lines 280-294) and in Figure S1.5 (lines 575-587). Obviously, the use of these new data is also reflected in several parts of the main text (e.g., *Introduction* section; lines 89-91, Fig. 1 caption; lines 485-493).*

Moreover, with the help of such data the statement in lines 66 to 67 should be substantiated, otherwise it remains a statement without a reference, which represents a certain disproportion to the well-prepared other evidences of all other changes (climate, Taxa community) represented in the area.

We added some references to the statement that our study region has experienced profound land-use changes linked to European settlement in the *Introduction* section (lines 69-72).

Apart from that I have a few little things to note:

1. *There is widespread knowledge that the Canadian forests are continually affected by fire, insects and disease. Thus, quite a lot of different natural disturbances are active. It should therefore become clear in the title as well as in the text (not only in line 37) that anthropogenic disturbances are meant here. This will also help to avoid misunderstandings regarding the cited literature. In other words, it will then become clear why literature that takes natural disturbances into account, such as that by Seidl et al. (2017), was not quoted.*

First, we changed the term “disturbance” to “anthropogenic disturbance” in the title and in *Abstract* (lines 21-24), to clarify that we are referring specifically to land-use related disturbance. Second, we added a whole paragraph discussing the potential role of changes to the natural disturbance regime caused by climate change (lines 150-163; with reference to recent literature; Seidl et al. 2017; *Nature Climate Change*, Sommerfeld et al. 2018; *Nature Communications*). In this new paragraph, we point out that in our study area, the impact of natural disturbances on compositional changes has most likely been small relative to the impact of anthropogenic disturbance, and in particular (citing relevant regional studies) that climate change has not increased the frequency of fire. In other words, our statistical signal of disturbance impacts is not an indirect consequence of climate change. Nevertheless, we used this enlargement of the discussion to comment on the broader, forward-looking context, noting “ongoing changes in disturbance regime that can be mediated by both land-use and climate change” (lines 167-168; 182-184). We think that these additions clarify and broaden the scope of the manuscript.

2. *In most Figures capital letters have been used to identify the individual Figures, but lower-case letters are indicated in the text of the Figure captions: that should become consistent.*

Capital letters in all figures have been replaced by lower-case letters.

Reviewer #3:

I have reviewed the paper "Dominant impact of disturbance relative to climate change on forest compositional changes over the past century in eastern Canada" by Dr. Danneyrolles and colleagues. This is an excellent piece of work on all levels and I very much support the conclusions. My only concern is to what extent is it novel compared to their previous papers on the study area and subject. I have read these papers but have not gone back to do a detailed analysis to see how this paper differs from the others. Maybe it is by investigating climate change as a driver. I do know that these papers contain information about forest dynamics and the disturbance, land-use drivers.

We appreciate these positive comments. In terms of novelty, our study goes beyond previous studies in several respects, which we hope to have communicated clearly in the revised manuscript. First, our study covers a broader region than previous papers, presenting a massive, mostly unpublished, database. Our data span temperate to boreal forests, and unlike most previous studies, we explicitly designed the analyses to disentangle effects of anthropogenic disturbance vs. climate change, thus providing a substantial advance in this important debate.

REVIEWERS' COMMENTS:

Reviewer #1 (Remarks to the Author):

This is the second time that I have reviewed the work by Danneyrolles and co-workers, now entitled "Anthropogenic disturbance is more important than climate change in explaining compositional changes in northern forests over the past century". The authors have considerably revised their manuscript, and have made many changes that contribute to an increased clarity of the text. Furthermore, they have addressed all my methodological concerns with extensive sensitivity analyses, which all underline the robustness of their results. Finally, the authors have changed the population-based index of human influence to a much more meaningful representation of land-use intensity. Overall, I'd like to commend the authors for their work – this is now a very nice manuscript that will certainly have a high impact in the community! I only have a few minor suggestions that could still be considered before the manuscript is handed over to the production department.

l27-30: The last sentence of the abstract is still weak. I'd suggest to work on this some more, and make a strong statement based on your findings that also non-experts can understand. This is important, as many people will (unfortunately) only read the abstract of your work!

l50: European settlement only makes sense in the North American context, but the intro text up to this point has been general, without a spatial reference, so maybe rephrase to 19th century?

l92: I'm not sure about the "best available" and you don't give a reference for this claim, so maybe tone it down to something like "the latter being a good proxy for"

l149-150: This sentence is somewhat inconsistent, the first clause talks about moisture while the second talks about temperature (warming).

l157: indirect effects of climate change via increased natural disturbances.

l157: fire frequency has

l185: highlights

l242: we report only results for the median in the main text

Fig. 1d: Why is the grey vertical line missing in the distribution here? In general I really like these figures, showing both the spatial pattern as well as the distribution of change!

Reviewer #2 (Remarks to the Author):

My comments relate only to the changes in the revised manuscript "Anthropogenic disturbance is more important than climate change in explaining compositional changes in northern forests over the past century" made by Danneyrolles et al. that I suggested in the first review.

I have carefully read and reviewed the comments on my suggestions and the implementation of the changes and have no longer proposed any changes. All comments from my side have been taken up, appropriately implemented and clearly marked in the text. I very much welcome the new figure S1.5 because it completes the whole picture of available data and the citation of the two current references Seidl et al. 2017 and Sommerfeld et al. 2018.

Reviewer #3 (Remarks to the Author):

I think the authors have done a very good job revising the ms and have been highly responsive to the reviewers concerns. I now believe the paper is a novel contribution and an extension of their previous papers. I believe it is worthy of publications in Nat Comm

RESPONSE TO REVIEWERS

Reviewer #1 (Remarks to the Author):

This is the second time that I have reviewed the work by Danneyrolles and co-workers, now entitled “Anthropogenic disturbance is more important than climate change in explaining compositional changes in northern forests over the past century”. The authors have considerably revised their manuscript, and have made many changes that contribute to an increased clarity of the text. Furthermore, they have addressed all my methodological concerns with extensive sensitivity analyses, which all underline the robustness of their results. Finally, the authors have changed the population-based index of human influence to a much more meaningful representation of land-use intensity. Overall, I’d like to commend the authors for their work – this is now a very nice manuscript that will certainly have a high impact in the community! I only have a few minor suggestions that could still be considered before the manuscript is handed over to the production department.

We sincerely want to thank all the three reviewers for their feedback and their support, which we think significantly contributed the present manuscript.

127-30: The last sentence of the abstract is still weak. I’d suggest to work on this some more, and make a strong statement based on your findings that also non-experts can understand. This is important, as many people will (unfortunately) only read the abstract of your work!

We did our best to clarify and strengthen the two last sentences of the abstract (see lines 30-35).

150: European settlement only makes sense in the North American context, but the intro text up to this point has been general, without a spatial reference, so maybe rephrase to 19th century?

Done: Indeed, it’s much better to keep a general context at this step of the introduction (see lines 53-56).

192: I’m not sure about the “best available” and you don’t give a reference for this claim, so maybe tone it down to something like “the latter being a good proxy for”

Done (lines 93-95).

1149-150: This sentence is somewhat inconsistent; the first clause talks about moisture while the second talks about temperature (warming).

We clarified this sentence (lines 157-160).

1157: indirect effects of climate change via increased natural disturbances.

Done (line 165).

1157: fire frequency has

Done (line 166).

l185: highlights

Done (line 194).

l242: we report only results for the median in the main text

Done (line 251).

Fig. 1d: Why is the grey vertical line missing in the distribution here? In general, I really like these figures, showing both the spatial pattern as well as the distribution of change!

That is simply because the grey lines show the 0 value (i.e. no changes) for the Δ Temperature and Δ SPEI, while for the Δ Population all cells values are >0 (see Supplementary Figure 5 for more details).

Reviewer #2 (Remarks to the Author):

My comments relate only to the changes in the revised manuscript “Anthropogenic disturbance is more important than climate change in explaining compositional changes in northern forests over the past century” made by Danneyrolles et al. that I suggested in the first review. I have carefully read and reviewed the comments on my suggestions and the implementation of the changes and have no longer proposed any changes. All comments from my side have been taken up, appropriately implemented and clearly marked in the text. I very much welcome the new figure S1.5 because it completes the whole picture of available data and the citation of the two current references Seidl et al. 2017 and Sommerfeld et al. 2018.

We sincerely want to thank all the three reviewers for their feedback and their support, which we think significantly contributed to the present manuscript.

Reviewer #3 (Remarks to the Author):

I think the authors have done a very good job revising the ms. and have been highly responsive to the reviewer’s concerns. I now believe the paper is a novel contribution and an extension of their previous papers. I believe it is worthy of publications in Nat. Comm.

We sincerely want to thank all the three reviewers for their feedback and their support, which we think significantly contributed to the present manuscript.